# Integrated evaluation of lung disease in single animals

**Pratyusha Mandal**[1][☯]*, **John D. Lyons**[2][☯]*, **Eileen M. Burd**[3], **Michael Koval**[4], **Edward S. Mocarski**[1]*, **Craig M. Coopersmith**[2]*

**1** Department of Microbiology and Immunology, Emory Vaccine Center, Emory University School of Medicine, Atlanta, GA, United States of America, **2** Department of Surgery, Emory Critical Care Center, Emory University School of Medicine, Atlanta, GA, United States of America, **3** Department of Pathology and Laboratory Medicine, Emory University School of Medicine, Atlanta, GA, United States of America, **4** Division of Pulmonary, Allergy, Critical Care and Sleep Medicine, Department of Medicine and Department of Cell Biology, Emory University School of Medicine, Atlanta, GA, United States of America

☯ These authors contributed equally to this work.
* mocarski@emory.edu (ESM); cmcoop3@emory.edu (CMC); jdlyons@emory.edu (JDL); pratyusha.mandal@emory.edu (PM)

## Abstract

During infectious disease, pathogen load drives inflammation and immune response that together contribute to tissue injury often resulting in organ dysfunction. Pulmonary failure in SARS-CoV2-infected hospitalized COVID-19 patients is one such prominent example. Intervention strategies require characterization of the host-pathogen interaction by accurately assessing all of the above-mentioned disease parameters. To study infection in intact mammals, mice are often used as essential genetic models. Due to humane concerns, there is a constant unmet demand to develop studies that reduce the number of mice utilized while generating objective data. Here, we describe an integrated method of evaluating lung inflammation in mice infected with *Pseudomonas aeruginosa* or murine gammaherpesvirus (MHV)-68. This method conserves animal resources while permitting evaluation of disease mechanisms in both infection settings. Lungs from a single euthanized mouse were used for two purposes-biological assays to determine inflammation and infection load, as well as histology to evaluate tissue architecture. For this concurrent assessment of multiple parameters from a single euthanized mouse, we limit *in-situ* formalin fixation to the right lung of the cadaver. The unfixed left lung is collected immediately and divided into several segments for biological assays including determination of pathogen titer, assessment of infection-driven cytokine levels and appearance of cell death markers. *In situ* fixed right lung was then processed for histological determination of tissue injury and confirmation of infection-driven cell death patterns. This method reduces overall animal use and minimizes inter-animal variability that results from sacrificing different animals for different types of assays. The technique can be applied to any lung disease study in mice or other mammals.

**Data Availability Statement:** All relevant data are within the paper and its Supporting Information files.

**Funding:** National Institute of General Medical Sciences (NIGMS) R01-GM072808,

T32GM095442 to CMC; NIGMS F32-GM117895 to JDL, National Institute on Alcohol Abuse and Alcoholism (NIAAA) R01-AA025854 to MKH; National Institute of Allergy and Infectious Disease (NIAID) R01-AI020211 to ESM and R21-AI142507 to ESM, CF@LANTA Director's Fund to PM.

**Competing interests:** NO authors have competing interests.

## Introduction

Animals models are essential for medical research investigating development, homeostasis, immunity, as well as disease mechanisms. The National Institutes of Health continues to stress the requirement for humane experimental strategies in animal research that align with the 3R principle: replacement, reduction and refinement [1]. Live animals are necessary tools to study mammalian processes at an organismal level [2], rendering replacement with alternative approaches non-viable option in many studies. Therefore, techniques that lead to accurate and objective information while using a minimal number of experimental animals are desirable. Here, we utilize intratracheal infection of *Pseudomonas aeruginosa* or intranasal infection of murine gammahespesvirus (MHV)68 as models for acute pulmonary inflammatory disease to demonstrate an integrated lung isolation technique. This method yields both histopathological and biological data from the same euthanized mouse.

Pulmonary diseases are some of the most significant human health hazards globally [3–5]. Changes in inflammatory signaling underlie many lung disease processes including asthma, chronic obstructive pulmonary disorder, acute respiratory distress syndrome, lung fibrosis, and cystic fibrosis [6–11]. Mice are widely used to investigate lung pathologies and the critical parameters of illness due to ease of genetic manipulation [12–14]. For respiratory infectious diseases, the primary determinants of outcome are a) pathogen load, b) infection-triggered inflammatory signaling, c) immune response and d) tissue injury. Even though an interconnected combination of these factors is a recognized driver of organ damage [4], the relative contribution of each often is not distinguishably resolved. Understanding disease processes requires careful assessment of each parameter. While quantification of infection, immune response, and inflammatory signaling may require fresh tissues, assessment of tissue morphology primarily relies on histology. When lungs are harvested without *in situ* fixation, the alveoli collapse, giving a deflated appearance that no longer preserves the original tissue architecture. *In situ* fixation by passing formalin through the trachea fixes all lung tissue and does not allow collection of fresh tissues from the euthanized mouse. Due to these issues, researchers use different groups of mice for generation of fixed and fresh lung tissue samples. This practice not only increases the total number of experimental animals but introduces complexities associated with animal-to-animal variability. If different mice are used to generate a/b/c and d, the correlation between obtained biological and histopathological information becomes difficult to objectively correlate. To address these issues, fresh and fixed tissue samples should be collected from the same mouse. Here, we utilize an integrated lung isolation technique following infection with *Pseudomonas aeruginosa* or MHV68 to demonstrate a method to achieve this goal.

*Pseudomonas aeruginosa* is an opportunistic pathogen that often underlies lung pathology in immunocompromised human beings such as patients who are critically ill due to other unrelated infections or people suffering from cystic fibrosis [15, 16]. Intratracheal inoculation of *Pseudomonas* is an established method infecting the lungs. Bacterium drives pulmonary inflammation and triggers injury via different processes including cytokine/chemokine production, as well as activation of cell death pathways [17–19]. This model has identified critical genetic factors, as well as inflammatory and immune determinants of bacteria-triggered acute lung pathology. In wild type (WT) mice, inoculation of *Pseudomonas* triggers pneumonia and sepsis [12, 18]. We isolated lungs for biological assays and histology at 12- and 24-hours post infection (hpi) with bacteria. MHV68 infection of mice is a model to study the contribution from gammaherpesviruses during inflammatory diseases [20]. Virus replication is suppressed by interferon gamma (IFNγ) such that IFNγ receptor knock-out (*Ifngr*$^{-/-}$) mice fail to control persistent virus. Intranasal infection of these mice results in chronic inflammation and

sustained injury of the lung. This is an established experimental model to assess the contribution from gammaherpesviruses to idiopathic pulmonary fibrosis [13]. Utilizing this model, we isolated lungs from WT and *Ifngr*-/- mice at 4 days(d) pi. For either bacterial or viral infection models, we perfused the right lung of the cadaver with formalin *in situ* for 20 minutes. The unfixed left lung was immediately excised. Segments from this lung was used for the determination of pathogen titer, cytokine levels and cell death markers. Fixed right lung tissues were used for histology. *In situ* formalin fixation preserved pulmonary architecture and produced histopathology images from undistorted lungs. In *Pseudomonas*-infected mice, this permits for accurate comparisons of hallmark bacteria-triggered changes in alveolar architecture between groups without artifacts due to the tissue isolation method. Our technique of lung isolation can be readily adapted to other pulmonary disease settings irrespective of infectious agent or animal species [21, 22].

## Results

### In situ formalin fixation of right lung

This fixation method prevents alveolar air sacs from collapsing and maintains the tissue resident lung architecture for analysis [23]. For *in situ* fixation, lungs need to be infused with 10% normal buffered formalin (NBF) before excision. We euthanized *Pseudomonas aeruginosa*-infected C57BL6/J WT mice undergoing pneumonia-induced sepsis [18] at 12 hours post infection (hpi) by IACUC-approved carbon dioxide inhalation. Using sterile technique, we immediately incised the neck to expose the trachea and inserted an angiocatheter into the tracheal lumen, directed two to three mm down the trachea caudally from the incision point (Fig 1A). We passed a 4–0 silk suture (surgical tie) circumferentially around the trachea, below the catheter insertion. The tied silk suture secured the catheter in place (Fig 1B). We then carefully incised along the midline of the chest wall such that underlying thoracic organs were not injured. Portions of the anterior chest wall were excised to fully expose both lungs. We tied a sterile 4–0 silk suture around the hilum of the left lung (Fig 1C left and zoomed in right panels) to prevent formalin flow into the left lung. The tracheal angiocatheter was connected to IV tubing attached to a 10 ml syringe fixed at a height of 20 cm above the cadaver (Fig 1D). This process also elevated the trachea above the level of the lung facilitating flow of formalin. We added 10 ml of NBF to the syringe that then flowed into the right lung. Formalin-induced inflation was observed in the right lung only confirming fixative flow was restricted primarily to this side (Fig 1E). We allowed the right lung tissue to fix for 20 minutes before being excised and placed into 10 ml formalin for further fixation (24–48 h). By maintaining multiple syringes attached to a stand at 20 cm above the workbench, we were able to fix the lungs of multiple mice at the same time. This minimized harvest time for researchers. 24 to 48 h later, fixed right lungs were put in histology cassettes and submitted for tissue sectioning. Paraffin-embedded unstained slides (for immunohistochemistry [IHC]) and hematoxylin-eosin (H&E)-stained slides (for histopathological analysis) were prepared by the Emory Histology Core at Yerkes National Primate Research Center. Unstained sections were used to detect appearance of apoptotic marker cleaved caspase-3 (Cl-CASP3), whereas H&E-stained sections were utilized in parallel to detect tissue morphology. During the 20 minutes of right-sided *in situ* fixation, we carefully removed the fresh left lung maintaining sterile techniques and cut it into three equal sections for different biological assays.

### Left lung occlusion

To determine whether the *in situ* method adequately limited formalin flow to the right lung only, we repeated our approach using hematoxylin dye (Fig 2A). Within a minute of starting

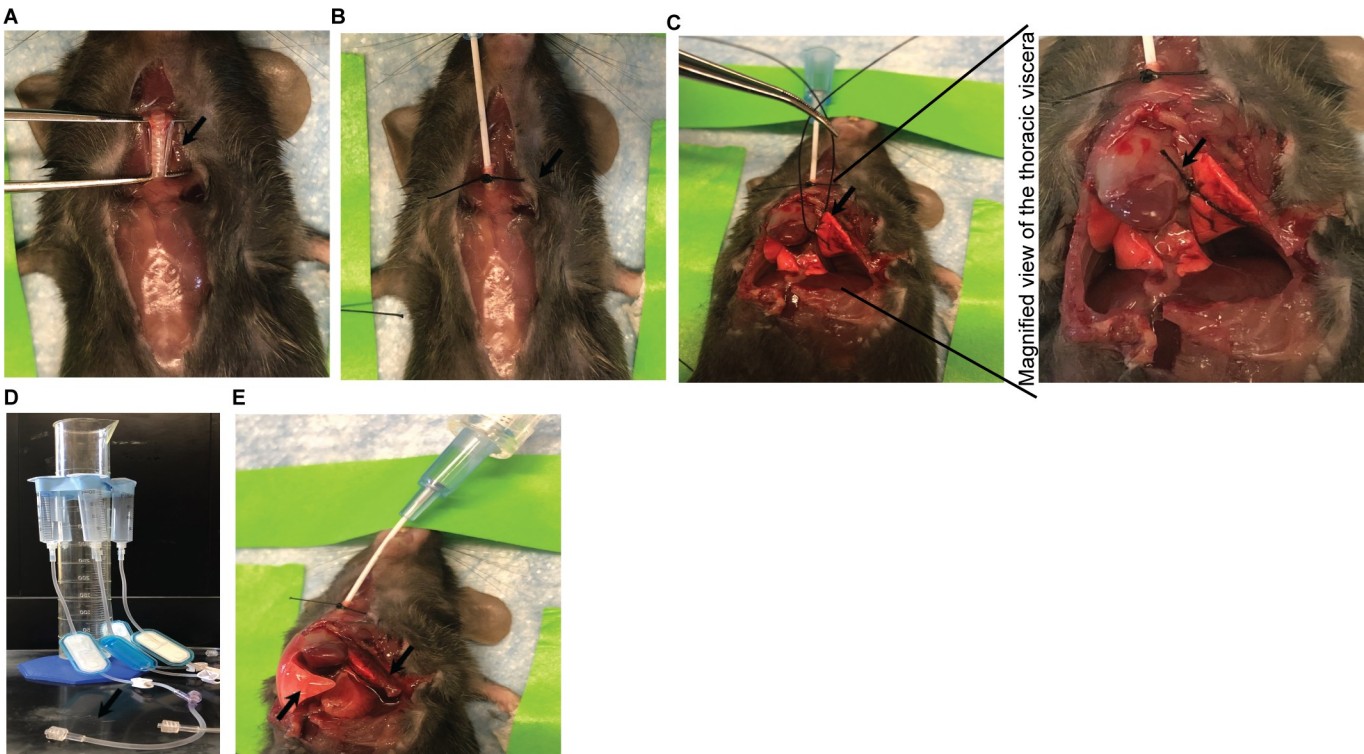

**Fig 1. Lung isolation method to obtain samples for biological and pathology assays.** The following steps after euthanasia are depicted: **(A)** trachea was dissected out using curved forceps, **(B)** angiocatheter was inserted into trachea and position fixed with black silk tie indicated by arrow, **(C)** a silk suture was placed circumferentially around the hilum of the left lung (left panel) to prevent flow of formalin from the tracheal angiocatheter into left lung tissue. Magnified view of the thoracic viscera (right panel) showing close-up of the suture (indicated by arrow) with respect to the lungs. **(D)** angiocatheter was connected to IV tubing attached to 10 ml syringes containing a fluid meniscus at 20 cm above cadaver as demonstrated, and **(E)** inflated right lung (indicated by left arrow) was perfused for 20 minute while left lung (indicated by right arrow) was excised for biological assays.

flow, the blue dye was visibly detected in right lung and not in the left lung. Continued observation confirmed flow to the right lung only, evidenced by progressively increased expansion of coloring in this tissue. At no point in time was dye observed in the left lung. Excised lung tissues were then scrutinized for presence of dye. Left lung exhibited no evidence of dye flow (Fig 2A end panel). Thus, our method directs flow of fluid to the right lung only, leaving the left tissues intact and available for additional biological assays. We also assessed viability of cells in the unfixed left lungs from WT mice (Fig 2B). Lungs from unmanipulated cadavers (Fig 2B, left panel) or cadavers undergoing *in situ* formalin fixation (Fig 2B, right panel) were excised. Single cell suspension of each lung was analyzed for viability using the trypan blue exclusion method. Approximately 90% of recovered left lung cells were viable (did not stain with trypan blue) independently of fixation conditions of the right side lung. This demonstrated that our harvest method did not cause loss of viability of cells in the unfixed tissues of the left lungs. Fixed right lungs retained a surprising, 70–80% viability (right panel), compared to the cells obtained from the non-perfused right side lungs (~90% viable). These data reveal that perfusion was sufficient to retain alveolar architecture despite *in situ* exposure to perfused 10% formalin. We next applied the *in situ* fixation technique to bacterial or viral infection models using *Pseudomonas* (Fig 3A–3D) or MHV68 inoculation respectively (Fig 3E–3H).

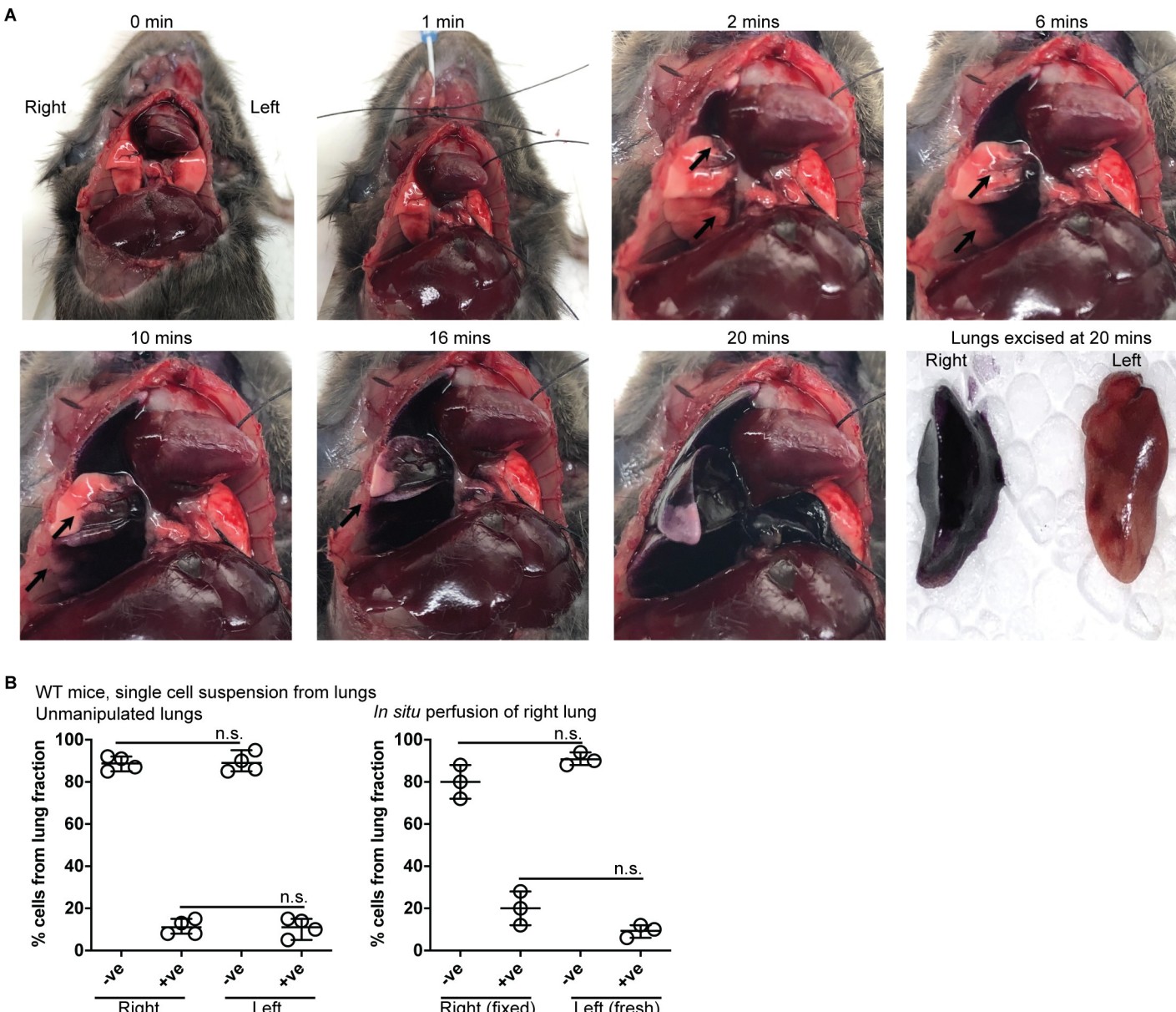

**Fig 2. Fluid flow to right lung and occlusion of the left lung.** (**A**) Hematoxylin dye was introduced to the right lung using *in situ* perfusion technique described in Fig 1. Pictures are from the same mouse (representative of three) depicting dye incorporation in the right lung over 0 through 20 minutes. By 20 minutes right lung turns dark blue while left lung does not show any visible dye. (**B**) Single cell suspensions obtained from the left lungs of uninfected mice were treated with typan blue for assessment of viability. Trypan blue negative (-ve) cells retain membrane property indicating viability, whereas positive (+ve) cells have membrane integrity damaged indicating loss of viability. Each data point represents one mouse with 3–4 mice per group. Error bars show mean error and range. Statistical analyses between groups were performed using unpaired t-test with Welch's correction. n.s. is non-significant.

## Pseudomonas and MHV68 titer

To evaluate *Pseudomonas* or MHV68 titer, we collected the lowermost section (nearest to the diaphragm) of the left lung from WT mice (for bacteria, 12 hpi), or WT and *Ifngr*$^{-/-}$ mice (for virus, 4 dpi) in sterile complete medium (Fig 3A and 3E). Equal amounts (by weight) of this tissue were sonicated in medium on ice and serially diluted to determine the infectious titer of either pathogen. We performed all biological assays with equal weight:volume ratio of tissue:

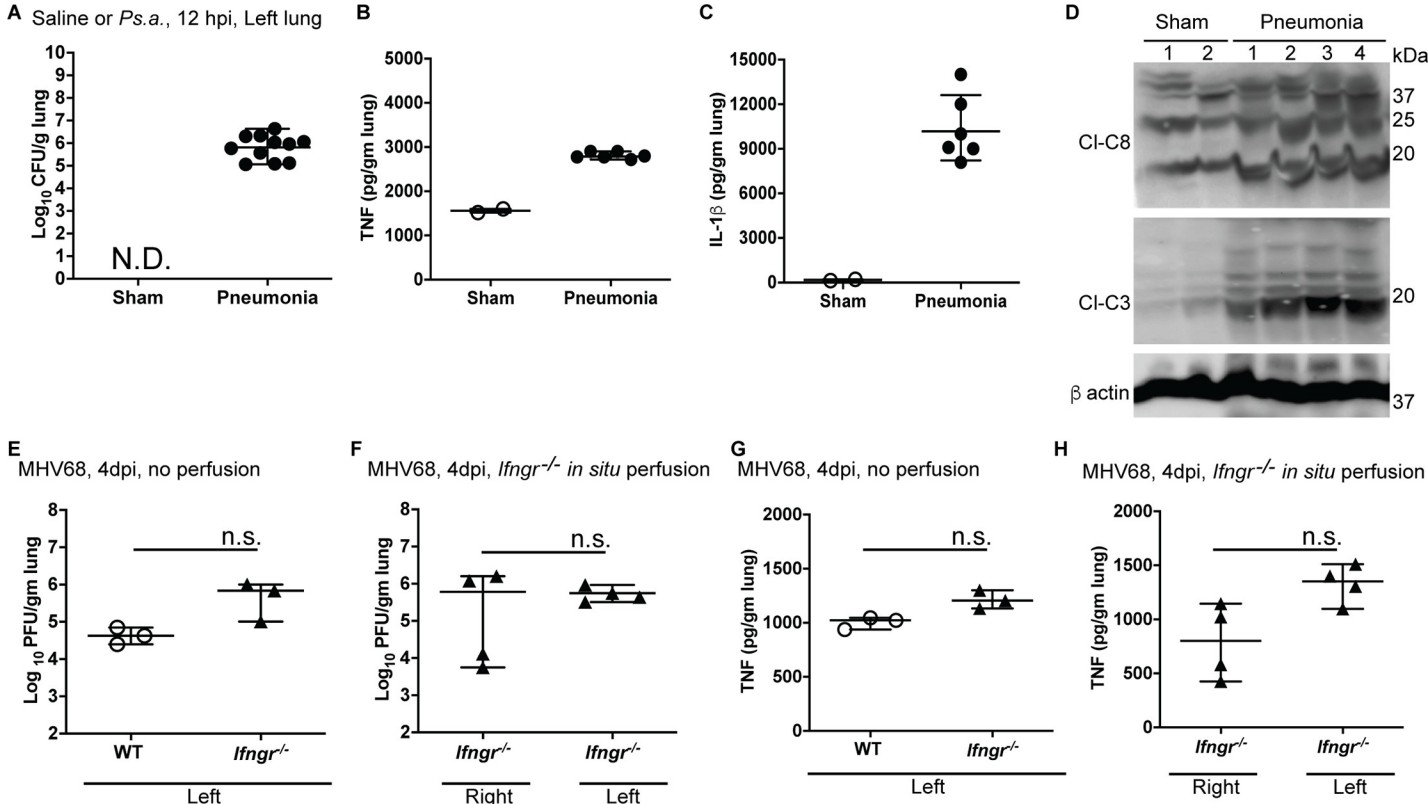

**Fig 3. Biological data from one mouse. (A, B)** Sham-treated or *Pseudomonas*-infected mice at 12 hours post infection (hpi; mice undergoing pneumonia) or treatment where each mouse was inoculated with 40 ul of $2X10^8$ CFU/ml *Pseudomonas aeruginosa* (ATCC 27853; approximately $8X10^6$ CFU per mouse) were used to quantify bacterial load (**A**, n = 2 and 11 mice for sham and infected groups respectively) on blood-agar plates and cytokines IL-1β (**B**), as well as TNF by ELISA (**C**; n = 2 and 6 mice for sham and infected groups respectively) by ELISA. N.D. is not detected. (**D**) Immunoblot of lung lysates from sham treated (n = 2) or septic (n = 4) WT mice indicating appearance of cleaved CASP8 (Cl-C8; 18 kDa) and Cl-CASP3 (Cl-C3; 19, 17 kDa) with loading control β-actin. (**E-F**) MHV68 titer in lungs from infected WT and *Ifngr*[-/-] mice without perfusion (**E**) or infected *Ifngr*[-/-] mice with *in situ* perfusion (**F**) 4 days-post-infection (dpi) where each mouse was inoculated with $5X10^5$ PFU of virus in 20 μl complete medium. (**G, H**) TNF levels detected by ELISA in lung sections from same mice depicted in (**E**) and (**F**) respectively. Each data point represents one mouse with 3–4 mice per group per condition. For each experiment all groups included comparable numbers of age-matched male and female mice. Error bars show mean error and range. Statistical analyses between groups were performed using unpaired t-test with Welch's correction. n.s. is non-significant.

solvent for titer, cytokine analysis or immunoblot. To evaluate bacterial titer (Fig 3A), soni-cated lung lysates were serially diluted in PBS before spreading on pre-warmed (37°C) blood agar plates. Plates were incubated at 37°C for 18–24 h before assessment of bacterial colonies. Plates exhibiting between 10–100 colonies were considered. Counted colonies were expressed as bacterial load after adjustment for tissue weight and dilution (Fig 3A). All infected mice exhibited comparable bacterial titer in the left lung. We did not detect bacteria in sham-treated mice confirming sterile technique was maintained during harvest without cross-contamina-tion. For determination of MHV68 titer, sonicated lysates were serially diluted in medium and plated on murine fibroblast monolayers (Fig 3E and 3F). Seven days later, plates were stained and wells with 20–200 viral plaques were counted and adjusted for tissue weight before calcu-lating titers. At 4 dpi, WT and mutant mice had comparable viral titer in the left lung, assayed without perfusion (Fig 3E). To determine whether *in situ* fixation impacts viral titers, we assessed titers in lungs from cadavers where our perfusion method was applied (Fig 3F). Left lungs from perfused *Ifngr*[-/-] mice exhibited similar titers (~$10^6$ PFU/gm tissue) when com-pared to left lungs from the unmanipulated mice (Fig 3E and 3F). Perfused right lungs showed a range of viral titer (~$10^3$–$10^6$ PFU/ml) suggesting that formalin perfusion for 20 mins

impacted titer in some mice. These data set the stage to evaluate infection-triggered inflammatory singling in the lungs from each mouse.

## Cytokine analysis and immunoblot (IB)

Acute *Pseudomonas* infection drives inflammatory cytokines such as TNF and IL-1β [24]. MHV68 infection triggers inflammatory cytokines detectable in lungs by 4 dpi [25]. To evaluate the quantity of infection-induced cytokines, we utilized the middle section from the left lung. We sonicated this tissue from each mouse in HBSS (Sigma) reconstituted with protease inhibitor cocktail (Roche). We used lysate for each sample to quantify TNF and IL-1β (for *Pseudomonas*, Fig 3B and 3C) or TNF (for MHV68, Fig 3G and 3H) levels by ELISA. We used the remaining topmost section (nearest to the trachea) of the left lung for IB to determine the appearance of known *Pseudomonas*-associated [26–28] cell death markers (Fig 3D). For *Pseudomonas* infection settings, sham-treated mice expressed detectable TNF, but not IL-1β, in lungs (Fig 3B and 3C). *Pseudomonas* expresses pro-apoptotic proteins that trigger apoptotic cell death in infected cells and mice [29]. As expected, apoptosis executioner caspase (CASP)3 was markedly activated (cleaved to produce bioactive 19 kDa form) in all infected samples when compared to sham-treated samples (Fig 3D). CASP8, a mediator of extrinsic apoptosis was also significantly processed to yield active, cleaved forms (22 kDa and 18 kDa) along with intermediate form (43 kDa). In all MHV68 mice without manipulation of lungs, infection drove detectable levels of TNF (Fig 3G). In *Ifngr*$^{-/-}$ mice where right lungs were fixed and left lung unfixed, the left tissues exhibited cytokine levels comparable to that observed in unmanipulated mice (Fig 3H). Fixed right lung, as observed with viral titer, exhibited a broader variation of cytokine quantities. These data, together with titer data, demonstrate that in our described *in situ* fixation method produces the quality of fresh tissue necessary for biological assays.

## Histology

To evaluate whether biological markers of infection correlated with tissue histopathology exhibiting bacteria-dependent inflammatory signaling, we assessed for morphology, appearance of infiltrates and cell death marker cleaved CASP3 (Fig 4A). In this figure, the first and second columns represent right lung harvested from two individual mice, one without and one with *in situ* fixation (as indicated in figure). The third column represents one infected mouse harvested at the same time with *in situ* fixation of the right lung. For H&E sections three magnifications are shown (increasing magnifications from top to bottom). The flattened alveolar spaces lead to thickened septal wall appearance in sections obtained without *in situ* fixation of the right lung. This is a recognized artifact of technique when lungs are placed directly in formalin without prior perfusion [23]. In contrast, a representative image from the *in situ* fixed lungs (Fig 4A second column) demonstrate crisp septal division. In lung sections obtained from mice infected for 24 h (S1A Fig) lungs from both sham-treated (S1A Fig, upper panel) and infected mice (S1A Fig, lower panel) exhibited flattened alveolar space in absence of *in situ* fixation. This confirmed the necessity of perfusion to preserve parenchymal architecture. Representative lung from one infected mouse demonstrated the known appearance of *Pseudomonas*-triggered inflammatory infiltrates and significant elevation of cleaved CASP3 (Fig 4A, lowest row) in complete accordance with IB (Fig 3D).

Thus, we show that in murine studies of pulmonary infectious diseases, adjacent lung tissues from the same mouse can be utilized for different purposes. The described method preserves the quality of both fixed and fresh tissues. Cell death markers confirmed the consistency of data generated. The biological assays from the left lung can easily be modified to determine

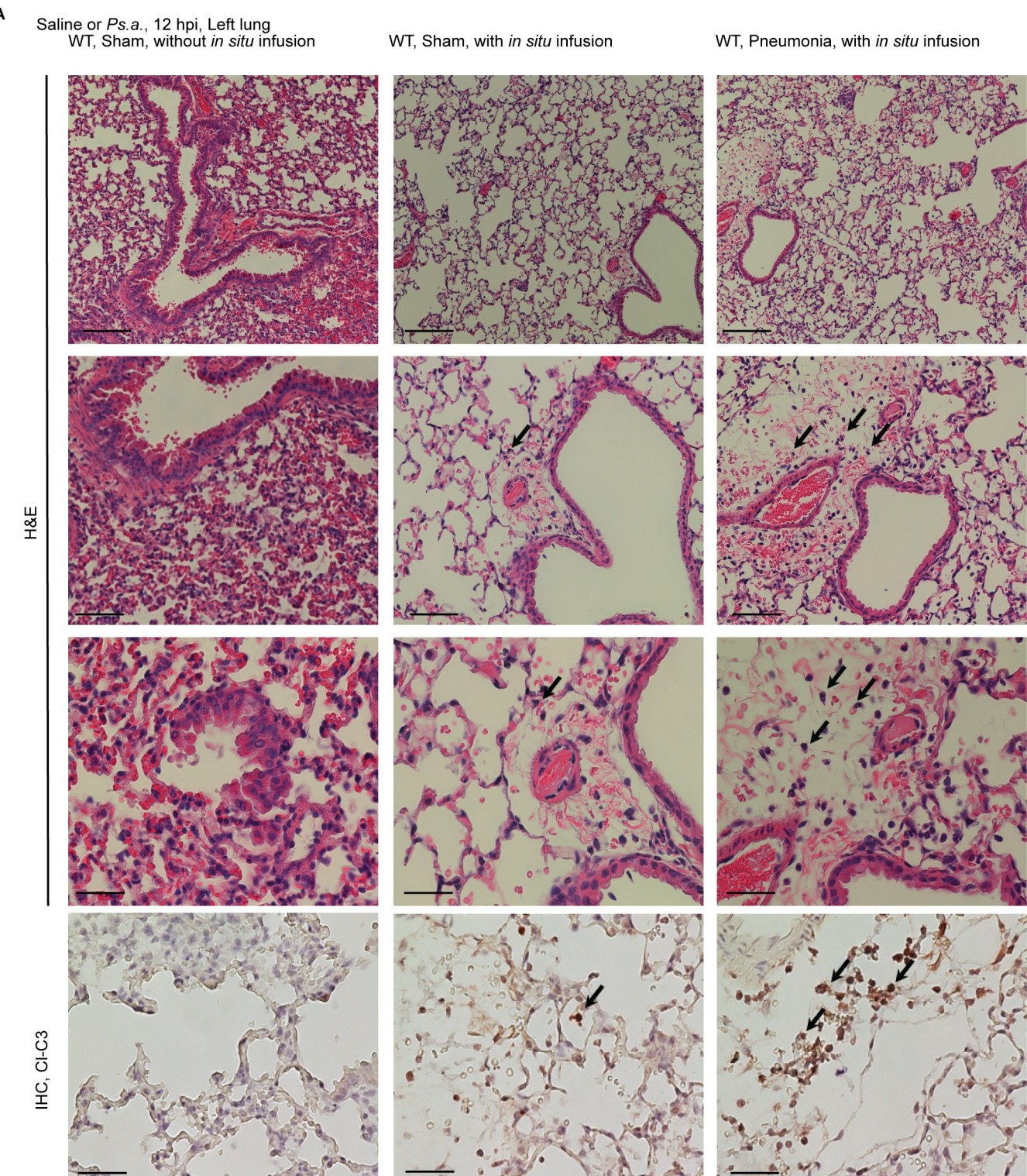

**Fig 4.** **(A)** Histology following H&E stain (upper panels, scale bar = 200 µm, 10X magnification of camera lens; middle panels, scale bar = 100 µm, 20X magnification; lower panels, scale bar = 40 µm, 40X magnification) and immunohistochemistry showing Cl-C3 *in situ* with hematoxylin counterstain (scale bar = 40 µm, 40X magnification) of the same lung segments from sham-treated or *Pseudomonas*-infected WT mice at 12 hours post (hp) treatment or infection (n = 1 representative image for each group). Left most H&E and IHC panels are from lungs without *in situ* fixation exhibiting thickened alveolar wall which is an artifact of the method. Arrows indicate representative inflammatory infiltration (in H&E section) or Cl-C3 positive cells (in IHC sections).

other disease markers such as immune cell infiltration (by flow cytometry), collagen deposition (by hyaluronidase assays) and pathogen DNA or gene expression patterns (by nucleic acid isolation) if required. Overall, we describe a lung isolation method that reduces total number of mice used for experiments by maximizing data generation. While we were surprised at the levels of viable cells and infectious virus remaining in lungs of perfused mice, our method refines lung isolation technique by overcoming fixation artifacts and simultaneously allowing generation of consistent biological data. Importantly, this method minimizes animal to animal variation by allowing all assays to be performed on individual mice. Therefore, here we describe a technique that enhances the 3R principle of animal research recommended by NIH.

## Methods

### Mice

Male and female C57BL/6J (JAX 000664), as well as C57BL/6J-background *Ifngr*$^{-/-}$ [13] mice were bred at Emory University. All infections were carried out with 8–12 weeks old mice where each experiment had comparable male and female mice. All animal experiments were conducted with approval according to the guidelines of the Emory University IACUC Animal Care and Welfare Review Committees.

### Intratracheal inoculation with *Pseudomonas aeruginosa*

*Pseudomonas aeruginosa* (ATCC 27853) stocks were maintained by a medical microbiologist (EB). Stocks were prepared in tryptic soy broth using rehydrated Culti-Loops® (Remel Microbiology Products, ThermoScientific Inc., Lenexa, KS). Stock quality was evaluated every month by subculture. Fresh subcultures were made from these plates daily for quality control procedures at the Emory Clinical Microbiology Laboratory. For experiments, a colony from a subculture plate was selected for inoculation in tryptic soy broth; bacteria were grown overnight at 37˚C. Pellet from centrifuged broth was resuspended in 0.85% saline (Remel Microbiology Products, ThermoScientific Inc., Lenexa, KS). 40 µl of inoculum from a stock of $2x10^8$ colony-forming units (CFU)/ml estimated by optical density at a wavelength of 600 nm. (approximately $8x10^6$ CFU/mouse) was inoculated in each mouse by intra-tracheal route as described before [12]. Briefly, mice were anesthetized with inhaled isoflurane (4% induction and 2% maintenance) and the trachea was exposed with dissection using sterile techniques. Inoculum was administered in the trachea using a fine-gauge needle. To ensure maximum flow of liquid inoculum, mice were held head-up for 10 to 15 seconds. For sham treatment, mice underwent anesthesia, midline cervical incision and injection of saline. The surgical incision was sealed using tissue glue and animals were euthanized by $CO_2$ inhalation following Emory University IACUC protocol.

### Intranasal infection with MHV68

Mice made unconscious by isoflurane inhalation were intranasally inoculated with $5X10^5$ PFU virus in 20 µl of media as described before [30]. Mice were maintained for 4 days before euthanasia.

### Hematoxylin staining of lungs

Cadavers of euthanized mice were set up for *in situ* perfusion of the right lung. 10 ml of hematoxylin (22110639, Fisher) was allowed to flow through the lungs for 20 minutes to detect leakage of dye into left lung.

## Lung cell isolation

Excised lungs were put in 1 ml ice cold medium and cut into ~3 mm pieces for isolation of cells as described before [2]. Lung fragments were digested with collagenase D (11088858001, Sigma; 1.5 mg/ml) in PBS, filtered through a metal sieve and subjected to erythrocyte lysis [31]. Viable cells were calculated using a hemocytometer and trypan blue exclusion.

## Viral and bacterial titer analysis

For organ titers, identical segments from the right lung of each mouse were collected in 1000 μl of complete medium. Complete medium is DMEM containing 4.5 g/ml glucose, 10% fetal bovine serum (F2442, Atlanta Biologicals), 2 mM L-glutamine (MT 25005CI, Fisher) with 100 units/ml penicillin and 100 units/ml streptomycin (MT 3002CI, Fisher). 100 mg of each lung segment was placed in 1000 μl complete medium, maintained on ice and homogenized using a Misonix Sonicator 2000 at program setting of two pulses for 10 seconds each at 15 Watts. For MHV68 infection settings, tissue lysates were diluted with seven serial ten-fold dilutions in media and second through seventh dilutions were plated on mouse fibroblast cells (NIH-3T3s, 200 μl/well). NIH-3T3 cells were plated 18 hours before experiment with $5 \times 10^5$ cell/well and maintained at 34°C incubator. After one hour adsorption with virus, cells in each well were covered with 5 ml warm carboxy methylcellulose as described before [30]. Plates were incubated for 6 days, stained with Giemsa and plaques were counted on day 7. Enumerated plaques were graphed as per gram of tissue. For *Pseudomonas* infection settings, 500 μg of lung sections was placed in 500 μl of media and homogenized. Lung lysates were diluted for five ten-fold serial dilutions in PBS and 200 μl from each sample was plated on pre-warmed blood agar plates for overnight incubation at 37°C. Dilution plates containing between 20–100 distinct colonies were considered for titer determination. Titers were expressed as bacteria for each lung as calculated using weight information.

## Lung lysate for cytokine ELISA

For cytokine ELISA, lung lysates were prepared from middle sections of left lung from each mouse. Comparable portions of lung section were weighed and sonicated in HBSS (Sigma) supplemented with protease inhibition cocktail (Roche; 1 tablet for 10 ml of HBSS). Approximately 0.5 mg of tissue was sonicated as described higher up in methods in 500 μl of supplemented HBSS. 75 μl from each lysate was used for TNF and IL-β murine cytokine ELISAs (R&D).

## Immunoblot

IB was performed as described before [32]. Briefly, top-most portion of left lungs were excised and placed in ice-cold HBSS. Equal weight of tissue (approximately 100 mg of tissue from each mouse) were sonicated in RIPA (25 mM Tris, 150 mM sodium chloride [Sigma], 1% NP-40 [Sigma], 1% sodium deoxycholate [Sigma], and 0.1% SDS [Sigma], pH 7.6; 200 μl per sample) supplemented with protease inhibitor and phosphatase inhibitor cocktail (Roche). After sonication, tissues were lysed on ice for 30 mins, collected by centrifugation at 150,000 rpm at 4°C for 20 mins using a Tomy TX-160 high speed refrigerated micro centrifuge. The resulting supernatant (tissue lysate) was collected for IB analysis. Beta actin levels were used to determine comparable protein quantities in all samples for IB. Proteins were transferred to PVDF membranes (Bio-Rad), treated with chemiluminescence reagent (Clarity, Bio-Rad) for signal development and imaged on Kwik Quant Gel Imaging System (Kindle Biosciences Inc).

Antibodies used were rabbit anti-cleaved Casp8 (8592, Cell Signaling Technology), rabbit anti-cleaved Casp3 (9661, Cell Signaling Technology) and mouse anti-β-actin (A2228, Sigma).

## Histology and immunohistochemistry

*In situ* fixed right lungs were excised and further fixed in ice-cold 10% normal buffered formalin (5700TS, Fisher) at 4˚C for 48 h and submitted for histology. For IHC detection, paraffin-embedded sections were prepared and stained as described [2] with rabbit anti-cleaved-CASP3 (1:100 dilution) at 4˚C, followed by biotinylated goat anti-rabbit secondary antibody (BA-1000, Vector Laboratories), streptavidin-horseradish peroxidase (HRP, SA-5004, Vector Laboratories) and peroxidase reaction reagent (Vector Laboratories). Slides were counter-stained using hematoxylin (22110639, Fisher) for 2 to 5 min and washed under tap water and finally with ultrapure water. Images were collected on a Nikon Elements microscope using Imaging Software-EIS Elements BR 3.10 (Nikon Instruments).

## Statistical analysis and reproducibility

Statistics shown on biological assays indicate mean error with range, and statistical comparison between groups were performed using unpaired t-test with Welch's correction in Graphpad Prism 8 (Graphpad Software Inc.). All graphs were graphed using same software. For biological assays multiple samples are shown to demonstrate reproducibility; for histology one representative image from each group is shown in the main figure (Fig 4) and 2–3 representative images are shown in S1 Fig. All analysis between groups were performed as

## Discussion

Pulmonary diseases, from chronic to acute, often mediated by pathogens are primary causes of death globally [3–5]. There is ongoing need to establish dependable models for pulmonary illnesses. For infectious lung pathologies, the complicated host-pathogen interaction that underlies disease requires *in vivo* assessment of genetic, inflammatory and immune determinants. All of studies demand live animals. Thus, researchers bear the responsibility of evaluating techniques that minimize number of animals utilized wherever possible without confounding quality or interpretation of the data. The approach described in this manuscript was developed utilizing acute bacterial and viral infections. Here, we have demonstrated that this method a) minimizes animal-to-animal variability, which should prevent associated errors in data interpretation, b) generates high quality, mutually consistent histopathological and biological evidence, as well as c) reduces the total number of animals needed to gather objective data.

Through *in situ* infusion, we generated fixed lung and fresh lung sections from individual animals. Perfusion followed by prolonged formalin fixation maintains tissue architecture. Without the first step, lungs exhibit tissue edema with increased alveolar septal width consistent with method artifact. As pneumonia or other pulmonary inflammatory conditions are known to cause tissue edema, such artifacts clearly confound observations. These investigations can be readily adopted for determination of genetic influences and efficacy of prevention strategies in other models. This modified fixation method cuts down on number of experimental animals used while retaining both quality and quantity of information necessary to understand the underlying disease process. One distinct caveat of this model includes the lack of ability to analyze bronchoalveolar-lavage (BAL) fluid. Due the restraint of fluid flow into the left lung, BAL (a flushed effluent of inflammatory intruding cell in the lung) cannot be obtained. Thus, separate groups of mice will be necessary for BAL. Additionally, distributions of infection and damage in lungs during inflammatory diseases are often not uniform. A prominent example is tuberculosis infection. It has been long recognized that in infected mammals apical and upper

lung sections are the primary sites of infection [33]. In such situations, or settings where the distribution of pathogen is not clearly understood, longitudinal sectioning (containing apical, mid and lower segments of the lung for each assay) of the fixed and unfixed tissue may be beneficial. Therefore, control experiments will be necessary for each experimental model to determine the differences in disease parameters in different sections of the lungs.

The ability to track individual animals for infection, inflammation and lung injury will significantly benefit studies examining the impact of antiviral and/or antibacterial therapy, as well as anti-inflammatory therapies, on disease outcome. In patients with acute respiratory distress, supportive care usually includes strategies to minimize inflammation related to both the underlying condition and associated with mechanical ventilation. During infectious pulmonary diseases, anti-inflammatory therapies are often combined with antibiotics or antivirals. This combined care regimen is intended to reduce the extent of tissue injury, minimize organ disfunction and prevent mortality [18, 34]. Even though such approaches are accepted, there is clear lack of comprehension into relative benefit of targeting infection versus inflammation. Likewise, the relative importance of specific components of the immune system such as eosinophils in asthma patients [35] have been difficult to assign. Our method will enable the targeting of different components of disease in order to reveal contributions, dependencies and overall outcomes. This insight into pathogen-host crosstalk is essential to understand the complex nature of pulmonary diseases. Undoubtedly, the strategies are applicable to any study using mice or other experimental animals beyond infection-driven acute pulmonary responses. Ultimately, the technique would allow users to employ the smallest number of animals for the determination of: a) the impact of a host modulation agent, b) the impact of an anti-viral or other anti-microbial (since changing pathogen burden will in turn alter host response and what is seen in the lungs) and c) the combination of both. This information will reveal whether host and pathogen modulations are additive, synergistic or paradoxically antagonistic for optimal intervention strategy in lung diseases.

## Supporting information

**S1 Fig.** (A) Histology following H&E stain (scale bar = 100 μm, 20X magnification on camera) from sham-treated (upper panel) or Pseudomonas-infected (lower panel) WT at 24 hours post (hp) treatment (t) or infection (i). Two representative images for sham and three presentative images for infected groups are shown.
(DOCX)

**S1 File.**
(DOCX)

## Acknowledgments

All contributors are included as authors.

## Author Contributions

**Conceptualization:** Pratyusha Mandal, John D. Lyons, Craig M. Coopersmith.

**Data curation:** Pratyusha Mandal, John D. Lyons.

**Formal analysis:** Pratyusha Mandal, John D. Lyons.

**Funding acquisition:** Edward S. Mocarski.

**Investigation:** Pratyusha Mandal, John D. Lyons.

**Methodology:** Pratyusha Mandal, Eileen M. Burd.

**Project administration:** Pratyusha Mandal, Edward S. Mocarski.

**Resources:** Pratyusha Mandal, Eileen M. Burd, Michael Koval.

**Software:** Pratyusha Mandal.

**Supervision:** Pratyusha Mandal, Michael Koval, Edward S. Mocarski, Craig M. Coopersmith.

**Validation:** Pratyusha Mandal.

**Visualization:** Pratyusha Mandal.

**Writing – original draft:** Pratyusha Mandal, John D. Lyons, Craig M. Coopersmith.

**Writing – review & editing:** Pratyusha Mandal, John D. Lyons, Craig M. Coopersmith.

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
