## [Decision Letter · Decision Letter 0]

22 Jul 2020

PONE-D-20-15152

Integrated Evaluation of Lung Disease in Single Animals

PLOS ONE

Dear Dr. Mocarski,

Thank you for submitting your manuscript to PLOS ONE. After careful consideration, we feel that it has merit but does not fully meet PLOS ONE’s publication criteria as it currently stands. Therefore, we invite you to submit a revised version of the manuscript that addresses the points raised during the review process.

ACADEMIC EDITOR: In addition to addressing the reviewers' concerns on the manuscript, the authors should discuss the limitations of the approach in this study. For example, various compartment of the lungs are not identical either functionally or structurally. Further, some infectious diseases such as TB has been documented to prefer the apical side of the lung. How these aspects would be addressed in your model system needs to be explained in the Discussion section.

We look forward to receiving your revised manuscript.

Kind regards,

Selvakumar Subbian, Ph.D.

Academic Editor

PLOS ONE

Journal Requirements:

Reviewers' comments:

Reviewer's Responses to Questions

**Comments to the Author**

1. Is the manuscript technically sound, and do the data support the conclusions?

Reviewer #1: Yes

Reviewer #2: No

2. Has the statistical analysis been performed appropriately and rigorously? 

Reviewer #1: Yes

Reviewer #2: I Don't Know

3. Have the authors made all data underlying the findings in their manuscript fully available?

Reviewer #1: Yes

Reviewer #2: Yes

4. Is the manuscript presented in an intelligible fashion and written in standard English?

Reviewer #1: Yes

Reviewer #2: No

5. Review Comments to the Author

Reviewer #1: Authors developed a nice technique to minimize the mouse usage in our in vivo experiments and clearly shown the difference between in situ fixation and normal fixation in their IHC studies. However authors mentioned that the in situ fixation process requires 20 minutes with flow through of 10 mL NBF, and authors failed to explain how they protect the other lung from flow through 10%NBF?. I would recommend the authors to include the text that address my concern.

Also I would recommend to show H&E staining of the whole lung (in low magnification) comparing in situ fixation and normal fixation, that will increase the strength of this manuscript

Reviewer #2: Summary:

The research article describes a protocol and its feasibility in different disease-based studies by minimizing the sacrifice of number of animals by using the same animal for histopathological and biological analysis. However authors have failed to justify the applicability of this potential protocol because of lack of experimental presentations. Manuscript has a lot of grammatical errors, some of which are mentioned in minor comments along with the major concerns below.

Major comments:

1. Authors mentioned- “During the 20 minutes of right-sided in situ fixation, we removed the left lung cut it into three equal sections and saved individually for biological assays”. Did authors check for any leakage of formalin in the left lung which was stitched to ensure tissue available for biological assays. There is no data presented in this context to show tissue was not or to what percentage it was affected by the formalin.

2. Authors mentioned-“We passed a 4-0 silk suture (surgical tie) circumferentially around the trachea, below the catheter insertion. The tied silk suture secured the catheter in place (Figure 1B)”. Did authors observe any back flow of formalin in the upper trachea

3. Did authors try single cell isolation from the left lung. Data on viability could be an interesting aspect to look into.

4. Since previous publications have already discussed about the in situ formalin fixation for 20 mins and its advantage over the other methods, authors here should have put more emphases and experimental justifications as to how a suture on the left lung was able to establish healthy tissue recovery for different experiments.

5. Authors mentioned about the applicability of this method in COVID-19 research as well. But they have not shown any data in that respect. Simple virus infection studies at BSL2 level along with the gram-negative bacteria reported here would have helped in justifying these statements.

Minor comments:

Please go through the typos etc in the manuscript some are mentioned below.

1. Line 123: “midline thought the chest”- correction needed.

2. Line 318: “lung injury will significant aid studies on following” correction needed.

3. Line 337: “both that might be result in results that” correction needed.

6. PLOS authors have the option to publish the peer review history of their article (what does this mean?). If published, this will include your full peer review and any attached files.

Reviewer #1: **Yes: **Murugesan Rajaram

Reviewer #2: No

---

## [Author Response · Author response to Decision Letter 0]

11 Dec 2020

ACADEMIC EDITOR: In addition to addressing the reviewers' concerns on the manuscript, the authors should discuss the limitations of the approach in this study. For example, various compartment of the lungs are not identical either functionally or structurally. Further, some infectious diseases such as TB has been documented to prefer the apical side of the lung. How these aspects would be addressed in your model system needs to be explained in the Discussion section.

Response: We appreciate the review and the editor’s note. We have modified text to clearly indicate how the lung structure varies, as well as limitations of our techniques. The modified Discussion now addresses the editor’s suggestions regarding the processes that take place in apical and basolateral epithelium of the lung.

1. Is the manuscript technically sound, and do the data support the conclusions?

Reviewer #1: Yes

Reviewer #2: No

Response: To address the concern by reviewer 2, we have revised the manuscript text so that conclusions are drawn from the data. Statistical comparisons are now included wherever applicable along with sample size information.

2. Has the statistical analysis been performed appropriately and rigorously?

Reviewer #1: Yes

Reviewer #2: I Don't Know

Response: In this modified manuscript, we added statistical comparisons wherever applicable along with sample size information.

3. Have the authors made all data underlying the findings in their manuscript fully available?

Reviewer #1: Yes

Reviewer #2: Yes

Response: We have included all data for all the figures as part of the manuscript.

4. Is the manuscript presented in an intelligible fashion and written in standard English?

Reviewer #1: Yes

Reviewer #2: No

Response: We have corrected all the errors.

5. Review Comments to the Author

Reviewer #1: Authors developed a nice technique to minimize the mouse usage in our in vivo experiments and clearly shown the difference between in situ fixation and normal fixation in their IHC studies. However authors mentioned that the in situ fixation process requires 20 minutes with flow through of 10 mL NBF, and authors failed to explain how they protect the other lung from flow through 10%NBF?. I would recommend the authors to include the text that address my concern.

Response: We have edited the text to clarify that unfixed left lung is excised and maintained cold for biological assays, immediately after tying the wire to prevent formalin flow into this lung. The new Figure 2 (an entirely new figure) includes images using hematoxylin dye (in place of formalin) demonstrating that in this occlusion method, the tie restricts flow to the left lung and dye only passes to the right lung.

Also I would recommend to show H&E staining of the whole lung (in low magnification) comparing in situ fixation and normal fixation, that will increase the strength of this manuscript

Response: We now include 10X, 20X and 40X magnification images of lung histology sections in Figure 4. Our initial submission only included 20X and 40X. The current range was selected to give a broad to magnified view of the lung that still demonstrated the differences of sections with or without in situ fixation. These distinctions are not objectively discernable at any magnification lower than 10X.

Reviewer #2: Summary:

The research article describes a protocol and its feasibility in different disease-based studies by minimizing the sacrifice of number of animals by using the same animal for histopathological and biological analysis. However authors have failed to justify the applicability of this potential protocol because of lack of experimental presentations. Manuscript has a lot of grammatical errors, some of which are mentioned in minor comments along with the major concerns below.

Response: We have modified the text and included new data in Figures 2, 3, 4 and S1 to address reviewer concerns. We have also corrected all grammatical errors. Please see answers to individual concerns below.

Major comments:

1. Authors mentioned- “During the 20 minutes of right-sided in situ fixation, we removed the left lung cut it into three equal sections and saved individually for biological assays”. Did authors check for any leakage of formalin in the left lung which was stitched to ensure tissue available for biological assays. There is no data presented in this context to show tissue was not or to what percentage it was affected by the formalin.

Response: New Figure 2 shows images using hematoxylin dye (in place of formalin) demonstrating that in this occlusion method, the tie restricts flow to the left lung and dye only passes to the right lung.

2. Authors mentioned-“We passed a 4-0 silk suture (surgical tie) circumferentially around the trachea, below the catheter insertion. The tied silk suture secured the catheter in place (Figure 1B)”. Did authors observe any back flow of formalin in the upper trachea

Response: A small amount of backflow of formalin to the upper trachea occurs as formalin flows with gravity. However, it is important to note that even though there is back flow to the trachea, formalin-fixation of the upper trachea does not influence analysis of lungs. 

3. Did authors try single cell isolation from the left lung. Data on viability could be an interesting aspect to look into.

Response: In new Figure 2, we show single cell viability counts. These data demonstrate that the unfixed left lung harvested in this way does not alter viability. Additionally, in Figure 3, we compare lungs excised from cadavers without manipulation of the lungs or with in situ perfusion to demonstrate that our perfusion technique does not alter MHV68 titers and cytokine levels. These data demonstrate that in situ formalin fixation of the right lung does not compromise outcome of biological assays using left lung.

4. Since previous publications have already discussed about the in situ formalin fixation for 20 mins and its advantage over the other methods, authors here should have put more emphases and experimental justifications as to how a suture on the left lung was able to establish healthy tissue recovery for different experiments.

Response: We thank the reviewer for this helpful comment. “In situ formalin fixation of right lung” section under Results now emphasizes that the suture is used to restrict formalin flow. The modified text is quoted below:

“We tied a sterile 4-0 silk suture around the hilum of the left lung (Figure 1C left and zoomed in right panels). This prevents formalin flow into the left lung. The tracheal angiocatheter was connected to IV tubing attached to a ten ml syringe fixed at a height of 20 cm above the cadaver (Figure 1D). This process also elevated the trachea above the level of the lung facilitating flow of formalin.”

Furthermore, “Left lung occlusion” section under Results describes in our observations confirming that the left lung is occluded in our technique. 

5. Authors mentioned about the applicability of this method in COVID-19 research as well. But they have not shown any data in that respect. Simple virus infection studies at BSL2 level along with the gram-negative bacteria reported here would have helped in justifying these statements.

Response: We have added the MHV68 data to Figure 3 demonstrating that our described technique can be adapted for lung studies with a BSL2 virus.

Minor comments:

Please go through the typos etc in the manuscript some are mentioned below.

1. Line 123: “midline thought the chest”- correction needed.

2. Line 318: “lung injury will significant aid studies on following” correction needed.

3. Line 337: “both that might be result in results that” correction needed.

Response: We have corrected these errors.

6. PLOS authors have the option to publish the peer review history of their article (what does this mean?). If published, this will include your full peer review and any attached files.

Do you want your identity to be public for this peer review? For information about this choice, including consent withdrawal, please see our Privacy Policy.

Reviewer #1: Yes: Murugesan Rajaram

Reviewer #2: No

---

## [Decision Letter · Decision Letter 1]

13 Jan 2021

PONE-D-20-15152R1

Integrated Evaluation of Lung Disease in Single Animals

PLOS ONE

Dear Dr. Mocarski,

Thank you for submitting your manuscript to PLOS ONE. After careful consideration, we feel that it has merit but does not fully meet PLOS ONE’s publication criteria as it currently stands. Therefore, we invite you to submit a revised version of the manuscript that addresses the points raised during the review process.

ACADEMIC EDITOR:

Figure 1: all panels should be labelled. Right now, the “zoom-out” image at far right doesn’t have any label. Correct the legend accordingly.

As commented by Reviewer#2, the legend for Fig4A and Fig S1A have the scale bars wrongly typed. Correct this information.

Line 470. What is 12hp ?. The manuscript quality would improve if such unusual abbreviations are avoided throughout.

We look forward to receiving your revised manuscript.

Kind regards,

Selvakumar Subbian, Ph.D.

Academic Editor

PLOS ONE

Reviewers' comments:

Reviewer's Responses to Questions

**Comments to the Author**

1. If the authors have adequately addressed your comments raised in a previous round of review and you feel that this manuscript is now acceptable for publication, you may indicate that here to bypass the “Comments to the Author” section, enter your conflict of interest statement in the “Confidential to Editor” section, and submit your "Accept" recommendation.

Reviewer #1: All comments have been addressed

Reviewer #3: All comments have been addressed

2. Is the manuscript technically sound, and do the data support the conclusions?

Reviewer #1: Yes

Reviewer #3: Yes

3. Has the statistical analysis been performed appropriately and rigorously? 

Reviewer #1: Yes

Reviewer #3: I Don't Know

4. Have the authors made all data underlying the findings in their manuscript fully available?

Reviewer #1: Yes

Reviewer #3: Yes

5. Is the manuscript presented in an intelligible fashion and written in standard English?

Reviewer #1: Yes

Reviewer #3: Yes

6. Review Comments to the Author

Reviewer #1: Authors carefully took actions to address all of my concerns and I am satisfied with the additional data provided by authors regarding my questions. Therefore I recommend editor to accept this manuscript for publication.

Reviewer #3: Authors addressed all questions raised. I have a few more concern.

Authors did not mention which statistical test they used for the data analysis in statistical analysis section.

There is typo in legend of the figure 4. Scale bar is given 200 mm, 100 mm and 40 mm for different panels. However, authors did not mentioned magnification of images respective to each panel in methodology section as well as in figure legend. I think scale bar should be presented in �m instead of mm.

7. PLOS authors have the option to publish the peer review history of their article (what does this mean?). If published, this will include your full peer review and any attached files.

Reviewer #1: No

Reviewer #3: No

---

## [Author Response · Author response to Decision Letter 1]

14 Jan 2021

ACADEMIC EDITOR: 

Figure 1: all panels should be labelled. Right now, the “zoom-out” image at far right doesn’t have any label. Correct the legend accordingly.

As commented by Reviewer#2, the legend for Fig4A and Fig S1A have the scale bars wrongly typed. Correct this information.

Line 470. What is 12hp ?. The manuscript quality would improve if such unusual abbreviations are avoided throughout.

Response: We have addressed all of the concerns in the manuscript. We have labeled and corrected corresponding legend for “zoom-out” image in Fig1. We have legends for Fig4A and FigS1A to indicate the correct measurement in �m We have corrected abbreviations and checked manuscripts to avoid unusual abbreviations. 

REVIEWER CONCERS:

Reviewer #3: Authors addressed all questions raised. I have a few more concern.

Authors did not mention which statistical test they used for the data analysis in statistical analysis section.

There is typo in legend of the figure 4. Scale bar is given 200 mm, 100 mm and 40 mm for different panels. However, authors did not mentioned magnification of images respective to each panel in methodology section as well as in figure legend. I think scale bar should be presented in �m instead of mm.

Response: We have now included in the statistical analysis section a complete description of the statistical test that was used for data analysis.

We thank the reviewer for catching the error in typing where scale bars are mentioned. We have corrected it for both Fig4 and FigS1A. We also indicate in both legends what lens magnification of the microscope used each scale bar corresponds to.

---

## [Editor Report · Decision Letter 2]

18 Jan 2021

Integrated Evaluation of Lung Disease in Single Animals

PONE-D-20-15152R2

Dear Dr. Mocarski,

We’re pleased to inform you that your manuscript has been judged scientifically suitable for publication and will be formally accepted for publication once it meets all outstanding technical requirements.

Kind regards,

Selvakumar Subbian, Ph.D.

Academic Editor

PLOS ONE
---

## [Editor Report · Acceptance letter]

29 Jun 2021

PONE-D-20-15152R2 

Integrated Evaluation of Lung Disease in Single Animals 

Dear Dr. Mocarski:

I'm pleased to inform you that your manuscript has been deemed suitable for publication in PLOS ONE. Congratulations! Your manuscript is now with our production department. 

Kind regards, 

on behalf of

Dr. Selvakumar Subbian 

Academic Editor

PLOS ONE